# The Absence of a Weak-Tie Effect When Predicting Large-Weight Links in Complex Networks

**DOI:** 10.3390/e25030422

**Published:** 2023-02-26

**Authors:** Chengjun Zhang, Qi Li, Yi Lei, Ming Qian, Xinyu Shen, Di Cheng, Wenbin Yu

**Affiliations:** 1School of Computer and Software, Nanjing University of Information Science and Technology, Nanjing 210044, China; 2Jiangsu Collaborative Innovation Center of Atmospheric Environment and Equipment Technology (CI-CAEET), Nanjing University of Information Science and Technology, Nanjing 210044, China; 3Jiangsu Engineering Center of Network Monitoring, Nanjing University of Information Science and Technology, Nanjing 210044, China

**Keywords:** link prediction, weighted networks, weak-ties theory, common neighbors

## Abstract

Link prediction is a hot issue in information filtering. Link prediction algorithms, based on local similarity indices, are widely used in many fields due to their high efficiency and high prediction accuracy. However, most existing link prediction algorithms are available for unweighted networks, and there are relatively few studies for weighted networks. In the previous studies on weighted networks, some scholars pointed out that links with small weights play a more important role in link prediction and emphasized that weak-ties theory has a significant impact on prediction accuracy. On this basis, we studied the edges with different weights, and we discovered that, for edges with large weights, this weak-ties theory actually does not work; Instead, the weak-ties theory works in the prediction of edges with small weights. Our discovery has instructive implications for link predictions in weighted networks.

## 1. Introduction

Complex networks play an increasingly important role in the era of information explosion, and the relationships between individuals can be simplified into the form of a network, which is a collection of nodes and edges. A typical network is composed of many nodes and some edges. The nodes are used to represent different individuals in the real system, and the edges are used to represent the relationship between individuals [1,2]. In 1961, Erdős and Rényi proposed the ER network, which features a low clustering coefficient, a short average distance, and a Poisson degree distribution [3,4]. In 1998, Watts and Strogatz proposed a network called the WS network, which has a high clustering coefficient, a small average distance, and a trend of exponential decay of the probability of large-degree nodes [5]. In 1999, Barabá and Albert proposed the BA network, which enjoys a small clustering coefficient and average distance [6,7]. The application of complex networks is reflected in all aspects of life. For instance, brain systems can be viewed as complex networks that can interact dynamically. Japp et al. studied the correlation between the structural properties of networks and the dynamic of networks, and they demonstrated that both functional and anatomical connectivity of the healthy brains share many features with small-world networks, but only to a limited extent, with scale-free networks [8]. Cats et al. developed a public transport robustness assessment model that computes ridership distribution and network performance metrics under planned capacity reductions [9].

Link prediction in the network refers to predicting the possibility of a connection between two nodes in the network that have not yet connected based on the network structure, such as their common neighbors [10]. Link prediction can be used to predict both unknown links (links that actually exist in the network, but which have not been detected), as well as future links (links that do not currently exist in the network, but which should exist or are likely to exist in the future) [11,12]. Research on link prediction can not only promote the theoretical development of network science and information science, but it also has great practical application value [13]. In recent years, with the rapid development of network science, its theoretical achievements have built a research platform for link prediction, making the research of link prediction closely related to the structure and evolution of the network. Therefore, the results of link prediction can be explained more theoretically [14,15,16,17].

Link prediction has very important practical implications and has been widely applied in many industries [18,19]. For example, link prediction can be used to predict potential academic partnerships [20,21], the number of citations of scientists [22] and the risk of diseases [23], recommend new friends/collaborators to people in social networks [24,25], the detection of anomalous emails [26,27], to analyze terrorist networks [28], and it can also be applied to the recommendation system to efficiently recommend suitable products to users [29,30]. Currently, link prediction research has attracted increasing attention from researchers [31]. Some scholars propose using a Markov chain for link prediction. In 2000, Sarukkai proposed and evaluated the notion of probabilistic link prediction and path analysis using Markov chains [32]. In 2002, Zhu et al. proposed a method for constructing a Markov model of a website based on past visitor behavior, and they used the Markov model for link prediction to help new users browse the site [33]. Additionally, some scholars propose using statistical relationships for link prediction. In 2002, Albert et al. proposed applying their statistical relational learning method to build a link prediction model [34]. In recent years, an increasing number of link prediction algorithms have been proposed. In 2006, Hasan et al. used supervised learning for link prediction [35,36], and machine learning methods have also been applied to link prediction [37,38,39]. Since traditional machine learning methods require massive preprocessing of data, Abbas et al. proposed a broad range of machine learning methods to predict the outcome of biomedical interactions and evaluated the performance of the traditional methods with more recent network-based methods [40]. Xian et al. presented a novel link prediction algorithm called GraphLP, which is based on network reconstruction theory [41]. Zhang et al. calculated the prediction accuracy of each edge in the network and found that the accuracy of edges with low edge betweenness is high, while the accuracy of edges with high edge betweenness is low. Therefore, they proposed an prediction algorithm, called RA-LP, to address this issue. This algorithm not only improves the accuracy of the edges with high edge betweenness, but it also improves the overall performance [42]. Shabaz et al. proposeed a method to predict future diseases based on existing health status using link prediction, and this explores how long the link survives [43]. In 2007, David and Jon developed a link prediction method based on analyzing the proximity measurement of nodes in the networks [44]. In 2008, Aaron Clauset et al. presented a general technique for inferring hierarchical structure from network data, and they further showed that knowledge of hierarchical structure can be used to predict missing connections in partially known networks with high accuracy [45,46].

In 2009, Guimer and Marta first mentioned the concept of a network error link, which means that the link that already exists in the network may be incorrect, and they presented a general mathematical and computational framework to address the problem of data reliability in complex networks [47,48]. In 2012, Almansoori et al. pointed out that prediction is one of the most appealing aspects in data mining, but none of the previous work in this domain has explored the prediction of links that may disappear in the future. Therefore, they proposed a link prediction model capable of predicting both links that might exist and links that may disappear in the future [49]. Recently, some scholars have used the link prediction method to infer important factors that have affected aviation network evolution. In the US aviation network, Liu et al. used the analysis of abnormal links in link prediction to find that, although some airports have a large number of common neighbors (simultaneously connected to two or more airports), there are no flights between them, and the reason is that they are limited by the factor of geographical location [50]. Inspired by link prediction, Wang et al. proposed a general framework for evaluating network evolution models, showing that link prediction has an important impact on revealing the network structure and evolution mechanism [51].

In terms of link prediction in weighted networks, some researchers have also conducted a series of studies. Murata and Moriyasu proposed three weighted similarity indices; they applied these indices to the networks of the *question-answer bulletin boards system*, and the results show that the prediction accuracy can be enhanced [52] with the consideration of weights. Lü et al. introduced these weighted similarity indices into a parameter α, and the parameter α was used to adjust the effect of the weight in the prediction. Link prediction in three real-world networks was carried out using this weighted index. As a result, they found that the optimal parameter α in the link prediction is mostly less than 0. This means that the edges with small weights play a more important role in link prediction [53,54]. On this basis, we found that, when predicting edges with large weights, the larger the parameter α is, the higher the accuracy of the prediction. On the other hand, when predicting edges with small weights, a smaller parameter α is better, as we can see in Figure 1.

## 2. Data and Methods

This section mainly introduces the data and link prediction methods used in our research. This research involves four undirected weighted empirical networks: (a) USAir: American Air Transportation Network. Each node in the network represents an airport, and, if there is a direct flight route between the two airports, then there is an edge between the two nodes corresponding to the two airports. The network contains a total of 332 airports and 2126 routes, and the frequency of flights between two airports is called the edge weight [55]. (b) Email: email communication network. The nodes in the network represent users. If there is an email exchange between two users, then an edge exists. The weight of the edge indicates the number of email exchanges between users, and there are 710 users and 5996 emails in this network [56]. (c) CGScience: computer geometry collaboration network. This network represents the cooperative relationship of scientists in the field of computational geometry and contains 3621 authors and 9461 cooperative relationships. Only two authors who have collaborated on articles will be connected. Therefore, the weight of the network is the number of papers the two authors have collaborated on [55]. (d) NetScience: network science collaboration network. This network is composed of scientists who have published papers. In this network, nodes represent scientists, and connected edges represent the cooperative relationship between scientists. It has 1589 nodes and 268 components, and we only consider the giant component in this paper, which contains 379 nodes and 914 connected edges. Obviously, the greater the number of collaborations between two scientists, the greater the weight of the connection between them [57]. If the connectivity of the network is not good, we use the giant component of the network to represent it. The characteristics of these networks are shown in Table 1.

As mentioned above, in a weighted network, the edges with large weights tend to cluster together. Similarly, the edges of small weights tend to cluster together. Among all the edges of a network, we take out a certain percentage of the edges, *P*, and we calculate the clustering coefficient of the network composed of *P*. After we sort the edges in ascending order according to weights, we remove the edges with the top proportion *P*, which can be regarded as the edges with small weights (SW). Similarly, after sorting the edges in descending order according to weights, we take the edges with the top proportion *P*, which can be regarded as the edges with large weights (LW). We randomly select the edges of the proportion *P* from all the edges, and this part of the edges can represent the edges of the weights of the entire network (RW). Finally, we calculate the clustering coefficient *C* of the network composed of the corresponding edges. We found that the clustering coefficient of the network composed of the edges with large weights and small weights is larger than the clustering coefficient of a network composed of random edges. That is, the edges with large weights tend to cluster together, and the edges with small weights also cluster together. The result showing this phenomenon is given in Figure 2. Among them, (a) represents the clustering coefficient of a network formed by edges with high and low weights is greater than that of a network formed by random edges in USAir network, (b) shows edges with high weights have a tendency to cluster with each other in Email network, (c) and (d) denote the same result in CGScience network and NetScience neteork. We can use visualization software to draw a network, such as the NetScience network, and the result showing this phenomenon is given in Figure 3. Among them, (a) represents the structure of the entire network, (b) shows the edges with small weights in the network, (c) denotes the connected edges with large weights, the number of their edges is all 5% of the entire network, and (d) represents randomly selected edges. We use precision to evaluate the performance of the link prediction algorithms [58]. Precision is defined after sorting the link prediction results. Define *l* as the number of accurately predicted edges and *L* as the the amount of predicted edges. If *l* of the first *L* edges belong to the test set, the precision can be defined as:(1)$$Precision=\frac{l}{L}$$

The four networks we use are undirected weighted networks. Define $S_{xy}$ as the similarity between node *x* and node *y*. The larger the value of $S_{xy}$ is, the more similar the node *x* and the node *y* is. In the experiment, the entire network was randomly divided into a training set $E^{T}$ and a test set $E^{P}$. The training set contains 90% of the entire network, and the test set contains 10% of the entire network. The test set is used to verify the accuracy of the prediction. To reduce the impact of randomness, the result is an average of 100 random divisions. We performed the experiment according to the following steps:

(i) First, we sort the links between the test set by their weight, and then we take 10% of the links with the largest weights from the sorted links and mark these links as $E^{LW}$. Similarly, we take 10% of the links with the smallest weights from the sorted links and mark these links as $E^{SW}$. Finally, we randomly take 10% of the links from the test set and note these links as $E^{RW}$. In the following, we study the precision for $E^{LW}$, $E^{SW}$ and $E^{RW}$.

(ii) We sort the links between the test set by their weight, and then we divide these links evenly into 10 groups according to their weights, namely, $E^{1}$, $E^{2}$…$E^{10}$, in which $E^{1}$ contains links with the smallest weight, and $E^{10}$ contains links with the largest weight. In the following, we study the best precision and the corresponding parameter α for these 10 groups.

## 3. Link Prediction Algorithms

Our research uses three similarity indices, the Common Neighbors (CN), Adamic-Adar (AA) index, and Resource Allocation (RA) index, whose definitions are as follows.

(i) CN. Let Γ(x) denote the neighbor of node *x* and Γ(y) denote the neighbor of node *y*; then, $S_{xy}$ can be expressed as the number of common neighbors of node x and node y, which can be expressed as follows:(2)$$S_{xy}=\left|\Gamma \left(x\right)\cap \Gamma \left(y\right)\right|$$
where |B| is the cardinality of the set *B*.

(ii) AA index. The idea of the algorithm is that the contribution of the common neighbor node with a high degree is less than that of the common neighbor node with a low degree [59], and $S_{xy}$ can be expressed as:(3)$$S_{xy}=\sum_{z\in \Gamma (x)\cap \Gamma (y)}\frac{1}{\log k(z)}$$
where k(z) is the degree of node *z*.

(iii) RA index. Some scholars were inspired by the process of resource allocation in the network, and they proposed a new similarity measure [17], which can be expressed as:(4)$$S_{xy}=\sum_{z\in \Gamma (x)\cap \Gamma (y)}\frac{1}{k(z)}$$

However, in the real world, most networks are weighted. For example, in airline networks, the distance between cities can be defined as the weight of this link; in social networks, the relationship between individuals is also weighted. Some scholars proposed a weighted similarity definition [52], so the weighted CN index, weighted AA index, and the weighted RA index (we regard them as WCN, WAA and WRA, respectively) can be expressed as:(5)$$S_{xy}=\sum_{z\in \Gamma (x)\cap \Gamma (y)}w\left(x,z\right)+w\left(z,y\right)$$
(6)$$S_{xy}=\sum_{z\in \Gamma (x)\cap \Gamma (y)}\frac{w(x,z)+w(z,y)}{\log(1+s(z))}$$
(7)$$S_{xy}=\sum_{z\in \Gamma (x)\cap \Gamma (y)}\frac{w(x,z)+w(z,y)}{s(z)}$$
where w(x,z) is the weight of the link between nodes *x* and *z*, $s\left(x\right)=\sum_{z\in \Gamma (x)}w(x,z)$.

A parameter α was introduced by researchers for these formulas [53], and α is used to control the relative contributions of weak ties to the similarity measures. The parameter-dependent indices for WCN, WAA, and WRA can be expressed as:(8)$$S_{xy}=\sum_{z\in \Gamma (x)\cap \Gamma (y)}{w(x,z)}^{\alpha }+{w(z,y)}^{\alpha }$$
(9)$$S_{xy}=\sum_{z\in \Gamma (x)\cap \Gamma (y)}\frac{{w(x,z)}^{\alpha }+{w(z,y)}^{\alpha }}{\log(1+s(z))}$$
(10)$$S_{xy}=\sum_{z\in \Gamma (x)\cap \Gamma (y)}\frac{{w(x,z)}^{\alpha }+{w(z,y)}^{\alpha }}{s(z)}$$
where $s\left(x\right)=\sum_{z\in \Gamma (x)}{w(x,z)}^{\alpha }$. Their research shows that the optimal value of α is mostly less than 1. That is, for some weighted networks, weak links play a more important role in link prediction. Our further research revealed an interesting conjecture, for edges with large weights in the test set, we should use a larger α to obtain a better prediction accuracy. For edges with small weights, we should use a smaller α to obtain a higher prediction accuracy. This phenomenon would have a great impact on the accuracy of weighted network link prediction.

## 4. Experiments and Results

Based on Formula (8), we first calculated the precision of $E^{RW}$, $E^{LW}$, and $E^{SW}$, with α ranging from −1 to 1, which aims to explore the different effects of weak-tie on predicting edges with small weights and large weights. The result is shown in the Figure 4. The precision of $E^{LW}$ (links with the larger weight) increases with the increase of α, while the precision of $E^{SW}$ (links with the smaller weight) increases with the decrease of α. For $E^{RW}$ (links are randomly picked from the test set), the best α normally ranges from −0.5 to 0. For example, (a) in Figure 4, for the USAir network, when predicting edges with large weights, with the parameter α ranging from −1 to 1, the prediction accuracy increases from 0 to 0.12, showing a significant positive correlation, and, when predicting small weight edges, as the parameter α increases from −1 to 1, the prediction accuracy drops from 0.03 to about 0.02, showing an obvious negative correlation effect, and while predicting random weight edges, when the prediction accuracy reaches the optimum the value is 0.06, the parameter α at this moment is around −0.5. In Figure 4b–d represent the same result that in order to achieve better prediction accuracy, it is advisable to use a smaller value of α for edges that have lower weights. We discovered that the optimal parameter value is negative, indicating that, in the context of link prediction, sometimes the ability of strong connections to promote connections is not as good as that of weak connections. That is, weak connections play a more important role than strong connections, which is the “weak connection” effect of link prediction. The results verified our conjecture: for links with a large weight, a larger α guarantees a high precision; for links with a small weight, a smaller α is more reasonable.

In the following, based on Formulas (8)–(10), we divided the links in the test set into 10 groups according to their weight, namely, $E^{1}$, $E^{2}$…$E^{10}$, in which $E^{1}$ contains links with the smallest weight, and $E^{10}$ contains links with the largest weight. Then, for each group $E^{n}$, we computed its best precision, as well as its corresponding α. Figure 5 shows the best precision of different groups $E^{n}$. We can clearly observe that, In Figure 5a,c, for the USAir network and the CGScience network, with the weight of the edges increasing, the accuracy results obtained by the two link prediction indices of CN and AA are almost the same, and the prediction trends of CN and the former two are also the same. For Email and NetScience networks, with the increase of edge weight, these three indices have almost the same accuracy to predict the trend, when CN increases, AA and RA increase; when CN decreases, AA and RA decrease simultaneously. Figure 6 shows the α corresponding to the best precision of different groups $E^{n}$. The green straight line indicates the RA index, the blue straight line represents the CN index, and the red straight line denotes the AA index. We select USAir, CGScience, Email, and NetScience as the experimental networks. As we can see, (a) in Figure 6, from $E^{1}$ to $E^{10}$, for the link prediction algorithm CN, RA and AA, with the increase of weight, the value of α also increases, and (b), (c), (d) have the same result that the value of α increases as the weight increases. which also verifies our assumption: for links with a small weight, we should use a small α to achieve a better precision, while for links with a large weight, a large α is required.

## 5. Summary and Discussion

In the past, most studies on link prediction focused on unweighted networks, and a great number of link prediction algorithms has been proposed. In recent years, some scholars have emphasized the important role of weak ties in link prediction [53], and some studies have developed the link prediction algorithms from unweighted networks of weighted networks. However, we believe that the weight of the links is as important as the prediction of existence of a missing link. Focusing on weak ties can significantly improve the prediction accuracy, which has certain implications for the development of link prediction. The weights of edges in the network are large or small, and overemphasizing edges with small weights will obviously ignore edges with large weights. Just because the weight of the network is large or small, we should still classify and explore the weight when we perform link prediction. Our research shows that, for links with different weights, the parameter α should be adjusted accordingly to achieve better precision. The results also show that, in link prediction for weighted networks, if we want to predict links with large weights, we should use a large parameter α, and if we want to predict links with small weights, we should use a small parameter α. By adjusting parameter α, our method enables the algorithms to work more accurately for links with different weights to improve the overall performance.

Our research discusses the impact of weights on prediction accuracy in detail. In a sense, it has a pivotal role in promoting the development of link prediction, and it may also provides a research direction for researchers. The conclusions drawn from experimentation on several empirical networks verified our research results, but we did not conduct a theoretical verification. We hope that scholars find inspiration on the basis of our current research and contribute further to the development of link prediction.

## Figures and Tables

**Figure 1 entropy-25-00422-f001:**
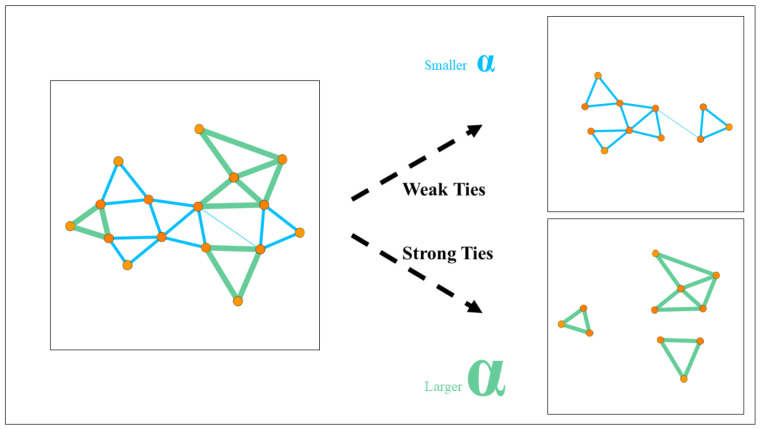
The graph on the left can be regarded as a weighted network, the edges with the thicker green line can be regarded as the edges with large weights, and the edges with the thinner blue line can be regarded as the edges with small weights. In a weighted network, edges with large weights usually gather together, and similarly, edges with small weights often gather together. For edges with small weights, we use a small α to adjust the effects of weights. For edges with large weights, we use a large α to adjust the effects of weights. Such a prediction is much more accurate than a prediction that uses the same α without distinguishing weights.

**Figure 2 entropy-25-00422-f002:**
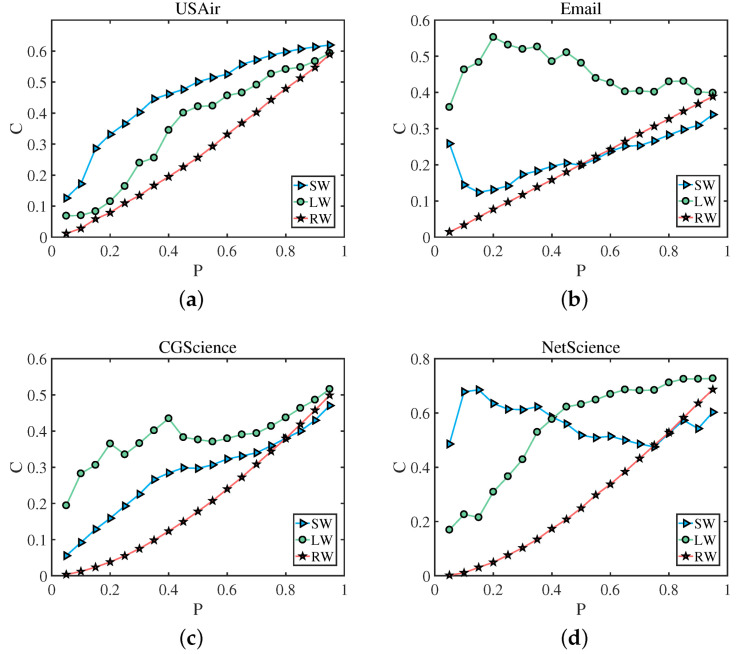
The horizontal axis is the edge of the entire network with a ratio of *P*, and the vertical axis is the average clustering coefficient (*C*) of the network composed of edges with the corresponding ratio of *P*. The average clustering coefficient of the network composed of edges with large weights (LW) and edges with small weights (SW) is basically greater than the average clustering coefficient of the network composed of edges with random weights (RW).

**Figure 3 entropy-25-00422-f003:**
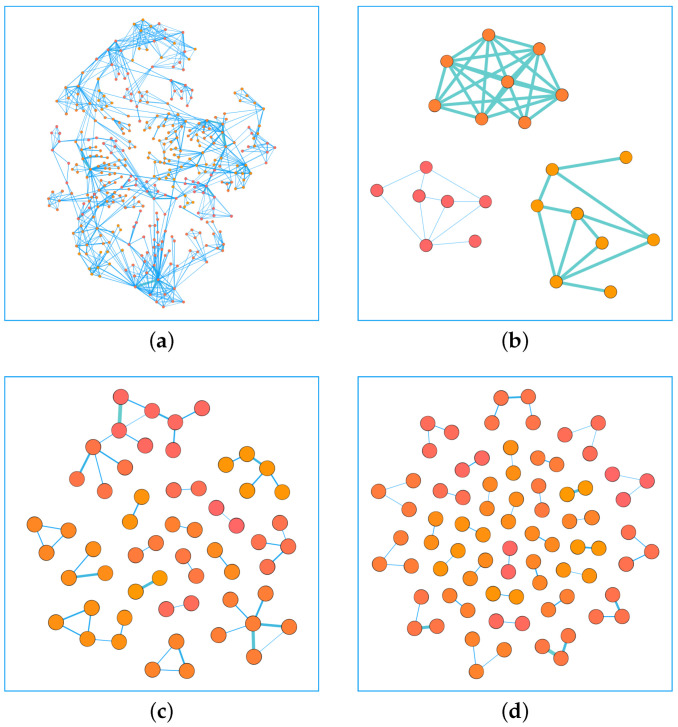
In the NetScience network, the entire network is shown in (**a**), 5% of the edges with small weights are shown in (**b**), 5% of the edges with large weights are shown in (**c**), and 5% of the edges are randomly selected as shown in (**d**). We can clearly see that the network composed of edges with large weights or edges with small weights has high clustering properties.

**Figure 4 entropy-25-00422-f004:**
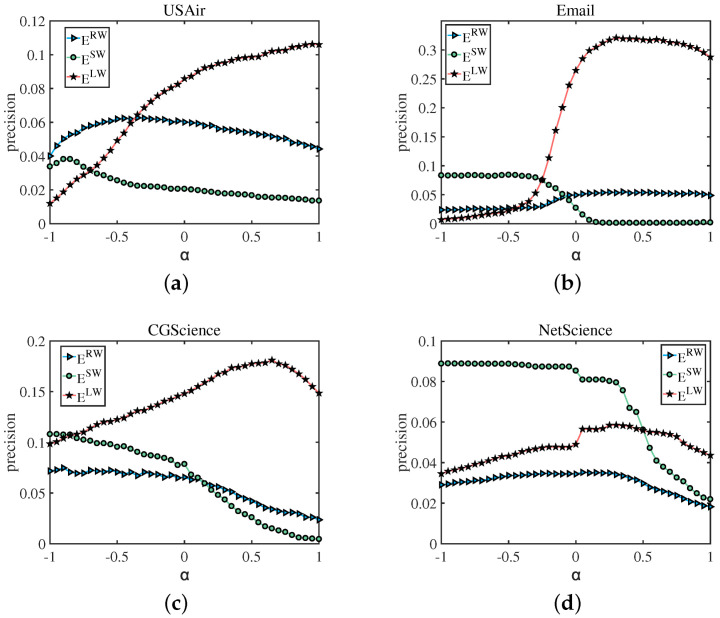
How the precision of links which are randomly picked from the test set ($E^{RW}$), links with large weight ($E^{LW}$) and links with small weight ($E^{SW}$) change with different α, and the range of numbers in bracket in the figure indicates the range of weights. For $E^{LW}$, the precision reaches the peak at a large α. For $E^{SW}$, the precision reaches the peak at a small α. For $E^{R}$, the best $E^{RW}$ is between −0.5 and 0.

**Figure 5 entropy-25-00422-f005:**
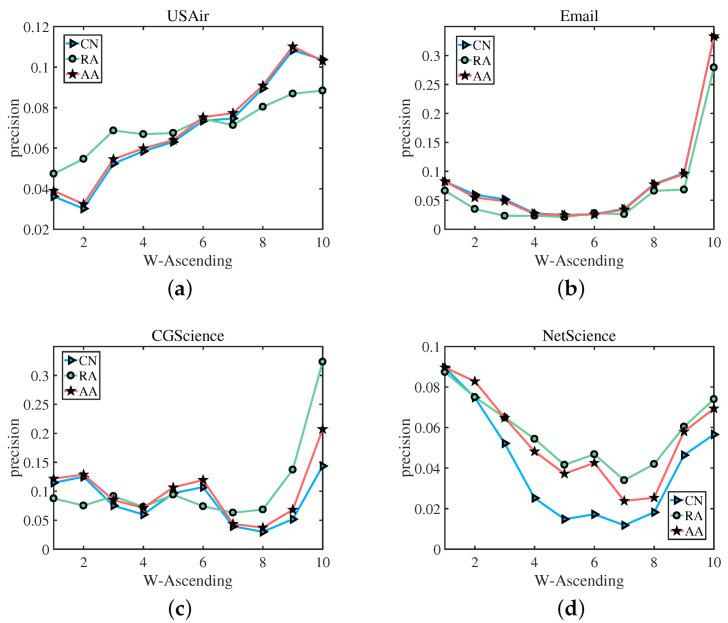
The best precision corresponds to a network composed of edges with different weight levels. The best precision was calculated using the CN, RA index, and AA index. W-Ascending means sort by weight in ascending order.

**Figure 6 entropy-25-00422-f006:**
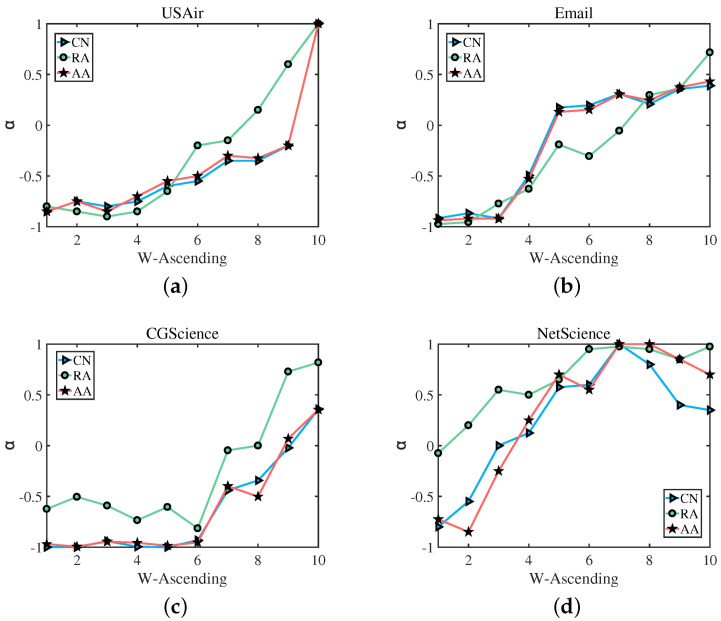
The α corresponds to the best precision of a network composed of edges with different weight levels. As the weight of links increases, the value of α also increases. The result shows that using a large α to predict edges with large weights will be more accurate, and using a small α to predict edges with small weights will be more accurate.

**Table 1 entropy-25-00422-t001:** Characteristics of the empirical network: number of nodes (*N*) and links (*M*), average degree (*K*), average shortest path length (*D*), clustering coefficient (*C*), degree assortativity (*R*).

Networks	*N*	*M*	*K*	*D*	*C*	*R*
USAir	332	2126	12.81	2.74	0.625	−0.208
Email	710	5996	16.89	3.00	0.409	−0.137
CGScience	3621	9461	5.23	5.32	0.540	0.168
NetScience	379	914	4.82	6.04	0.741	−0.082

## Data Availability

Not applicable.

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
