# Peer review of "The Absence of a Weak-Tie Effect When Predicting Large-Weight Links in Complex Networks"

_entropy, 2023, doi:10.3390/e25030422_

Round 1

Reviewer 1 Report

The paper presents a study on the weak ties in link prediction. It is interesting and can be considered for publication with appropriate revisions, as detailed below:

1.       Parameters should be defined when they are used for the first time.

2.       You take 10% of the test set and define ERW, ELW and ESW. But based on the definition of precision (1), how can you get these three different curves in Fig.4?

3.       What does the color of the nodes mean in Fig.3? 

4.       The contributions should be made in this research more clearly. For example, modification of the formulas with the alpha power is a contribution or existed before?

5.       Some grammar errors and typos should be fixed.

6.       There are format errors in the references; please correct them.

Reviewer 2 Report

Link prediction is a popular topic in information filtering. However, most existing algorithms are for unweighted networks and there are relatively limited studies on weighted networks. Previous studies on weighted networks have revealed the importance of small-weight links and the impact of the weak-ties theory on prediction accuracy. In this paper, the authors studied edges with different weights and found weak-ties theory works for edges with small weights but not for those of large weights, which has implications for link predictions in weighted networks. I have the following comment for the authors.

1. I suggest the authors add more discussion about recent research in the field.

2. The format of the Reference is incorrect; for instance, Reference 38 misses the page and volume information.

3. The Figures should be discussed in more detail. For instance, in figure 2 (b), when P is between 0.6 and 1, why SW is lower than RW?

4. In line 190, I recommend the authors modify the sentence as “Our further research revealed an interesting Conjecture,” , which is consistent with the following content.

5. There are some grammar mistakes and typos; please correct them.
